# A Review of the Ecological and Biogeographic Differences of Amazonian Floodplain Forests

**Florian Wittmann** [1,2,*]**, John Ethan Householder** [1]**, Maria Teresa Fernandez Piedade** [2]**, Jochen Schöngart** [2]**, Layon Oreste Demarchi** [2]**, Adriano Costa Quaresma** [1,2] **and Wolfgang J. Junk** [3]

1  Department of Wetland Ecology, Institute of Geography and Geoecology, Karlsruhe Institute of Technology, 76131 Karlsruhe, Germany
2  MAUA Working Group, National Institute of Amazonian Research (INPA), Manaus 69067-375, Brazil
3  Brazilian National Institute for Wetlands (INAU), Federal University of Mato Grosso, Cuiaba 78060-900, Brazil
*  Correspondence: florian.wittmann@kit.edu

**Abstract:** Amazonian floodplain forests along large rivers consist of two distinct floras that are traced to their differentiated sediment- and nutrient-rich (várzea) or sediment- and nutrient-poor (igapó) environments. While tree species in both ecosystems have adapted to seasonal floods that may last up to 270–300 days year$^{-1}$, ecosystem fertility, hydrogeomorphic disturbance regimes, water shortage and drought, fire, and even specific microclimates are distinct between both ecosystems and largely explain the differences in forest productivity and taxonomic composition and diversity. Here, we review existing knowledge about the influence of these environmental factors on the tree flora of both ecosystems, compare species composition and diversity between central Amazonian várzeas and igapós, and show that both ecosystems track distinct species life-history traits. The ecosystem-level and taxonomic differences also largely explain the biogeographic connections of várzeas and igapós to other Amazonian and extra-Amazonian ecosystems. We highlight the major evolutionary force of large-river wetlands for Amazonian tree diversity and explore the scenarios by which the large number of Amazonian floodplain specialist tree species might even contribute to the gamma diversity of the Amazon by generating new species. Finally, we call attention to the urgent need of an improved conservation of Amazonian várzea and igapó ecosystems and their tree species.

**Keywords:** anoxia; seed dispersal; life-history traits; forest succession; beta diversity; refugia



## 1. Introduction

Large-river wetlands cover approximately 750,000 km$^2$ (~11%) of the area of the Amazon basin, most of which is forested. These seasonally flooded forests are the most species-rich floodplain forests on Earth [1], store significant stocks of carbon [2,3], and provide a variety of valuable ecosystem services in terms of providing habitat, regulating biogeochemical cycles, and provisioning food and material goods for human welfare [4]. The fine-scale habitat heterogeneity of Amazonian floodplain forests has regional implications for the origin and maintenance of Amazonian biodiversity [5]. Floodplain forests also operate as habitat refugia by mitigating climatic stressors, because wetland habitats show reduced temperature and soil moisture variability relative to adjacent uplands i.e., [6,7]). In this sense, wetlands may allow for the survival of populations of species living in constantly changing climatic contexts [8].

Amazonian large-river floodplains are characterized by the contrasting fertility of their flood waters and alluvium. Sediment- and nutrient-rich white-water rivers form floodplains known as várzea. Sediment- and nutrient-poor black- and clear-water rivers form floodplains classified as igapó [4,9–11]. Prance [10] and Kubitzki [12] emphasized the floristic differences between várzea and igapó forests based on herbaria collections of characteristic indicator tree species. These publications were foundational for our understanding of the floristic differences and evolution of Amazonian floodplain forests.

While much of the knowledge summarized in these publications still holds true, over the last three decades, studies from diverse fields of geology, geomorphology, geography, botany, and ecology have helped to further our understanding of the environmental and floristic patterns of Amazonian floodplain forests and the underlying processes that sustain them.

Here, we review the existing literature on the environmental differences and biogeography of várzea and igapó tree communities. Our review focuses on the following questions: (1) What are the floristic differences between igapó and várzea forests? (2) What are the main environmental differences between these ecosystems? (3) Do the environmental differences favor specific strategies and/or life-history traits of colonizing tree species? (4) What are the mechanisms of the evolution of flood-adapted specialist tree species? By responding to these questions, we infer the differential evolution of Amazonian igapó and várzea forests, describe their biogeographic connections to other Amazonian and extra-Amazonian ecosystems, and show that Amazonian floodplain forests importantly contribute to the tree diversity of the Amazon basin.

## 2. The Physical Setting of Amazonian Floodplains

Wetlands cover an area of approximately 2.33 Mio km$^2$ (~34%) of the Amazon basin. Out of these, approximately 1.7 Mio km$^2$ are river wetlands, either as episodically flooded riparian areas along the headwaters of upper courses (approximately 1 Mio km$^2$) or as seasonally flooded large-river floodplains along the middle and lower courses (approximately 750,000 km$^2$, [4,13]. Because of seasonal rainfall linked to the yearly shift of the Intertropical Convergence Zone, combined with a flat landscape over most parts of the basin, Amazonian large-river wetlands are subject to a predictable flood pulse in magnitude and timing [14], with one high-water and one low-water period during the year. Seasonal water-level oscillations of major Amazonian rivers, such as the Solimões and lower Negro Rivers, amount to 9–12 m in the central Amazon basin, reaching maxima of up to 12–15 m at the middle to lower courses of the southern Amazon tributaries of the Madeira, Purús and Juruá Rivers [15]. In western and eastern Amazonia, flood pulses decrease to 3–6 m, either because of reduced catchment areas and higher slopes in the west or enlarged riverbeds in the east [16]. The flood pulse is considered the main environmental forcing factor for the biota of Amazonian large-river wetlands (Junk et al., 1989). Many wetland organisms require water-level fluctuations for the survival of their populations, including many fish and invertebrate species with spawning and feeding migrations between rivers and floodplains, i.e., [17–21]. Many aquatic and terrestrial mammals, birds, reptiles, and amphibians synchronize their life cycles with the hydrological cycle, i.e., [22–24]. Herbaceous and woody floodplain plant species possess a series of morphoanatomical and physiological adaptations to the seasonal inundations, i.e., [25–27].

The Amazonian drainage system is likely as old as the existence of neotropical rainforests, which established in large parts of northern South America since at least the Upper Eocene, approximately 50 Ma BP [28–30]. With the Andean uplift, it was profoundly re-shaped, mainly during the middle to upper Miocene, between 23 and 10 Ma BP [31]. During the Paleogene, Andean mountain building already generated a system of depressions in the Miocene foreland basin of the western Amazon, where vast freshwater wetland systems established, likely several times influenced by marine transgressions. At approximately 7 Ma BP, the Amazon River reversed to the East with the closure of the Vaupés Arch, and Andean sediments started to reach the Atlantic Ocean [32]. Western Amazonia developed into a landscape of widespread river terrace systems and entrenched rivers with a high sediment load [33]. Today, Andean sediment covers most of the area of the western Amazon, where the Caquetá-Japurá river to the north and the Madeira river to the south form the approximate natural boundaries between the Andean sediment in the west and older, cratonic geological formations of central and eastern Amazonia [34].

Amazonian large-river wetlands display a wide range of environmental settings depending on the geology and geomorphology of basins where rivers originate from.

The geochemical differences between white-, black-, and clear-water rivers were first described by Sioli [9]. Old cratonic formations, such as the Guiana and central Brazilian Shields in the N and S of the Amazon basin, are drained by sediment- and nutrient-poor black- and clear-water rivers, while the Andes and sub-Andean regions in NW, W, and SW Amazonia are drained by sediment- and nutrient-rich white-water rivers [9,11]. The ecosystems flooded by white-water rivers were classified as várzea, while those flooded by black- and clear-water rivers were classified as igapó [4,9,11]. Várzeas are formed along the main stem of the Amazon river and its large white-water tributaries Madeira, Purús, Caquetá-Japurá, Juruá, Jutaí, Javaris, and Putumayo-Içá. These rivers transport, deposit, and remobilize large amounts of sediment and dissolved matter from the Andes to the Atlantic Ocean. The sediment load contains large amounts of multilayered clay minerals with elevated cation exchange capacity, such as smectite, illite, and montmorillonite, that release nutrients during the weathering process, resulting in a relatively high fertility of alluvial substrates. The water is slightly acidic to neutral (pH 6–7) and is dominated by Ca, Mg, and carbonates [9]. Electric conductivity decreases from the west to the east and amounts to 140–120 $\mu$S cm$^{-1}$ near the Andes to 50–30 $\mu$S cm$^{-1}$ at the lower Amazon [35]. In contrast, igapó rivers drain old, strongly weathered Tertiary sediments of Paleozoic and pre-Cambrian origin. The water is poor in suspended solids, transparent, and of brown to blackish color when originating from forested regions, or greenish clear when originating from regions predominately covered by savannas, particularly in the eastern parts of the Amazon basin [16]. The most important black-water rivers are the Negro, Coari, and Uatumã rivers in the central Brazilian Amazon. The very low content of dissolved matter results in an electric conductivity of <20 $\mu$S cm$^{-1}$, and the waters are mostly acidic due to the high amounts of dissolved organic material, with a pH of 4–5. The content of alkali-earth metals is low and contributes less than 50% of the total cation content, in which sodium dominates, while the principal anions are Cl$^{-}$ and SO$_4^{2-}$ [36]. The most important clear-water rivers are the Tapajós, Xingu, and Tocantins-Araguaia originating from SE Amazonia, and the Branco, Trombetas, Paru, and Araguari rivers originating from the Guiana Shield. River waters and sediments vary in fertility but are relatively sediment- and nutrient-poor [4].

## 3. Floristic Differences between Várzea and Igapó

Using the data base of the Amazon Tree Diversity Network—ATDN, which consists of approximately 1500 floristic inventories across the Amazon basin and the Guiana Shield, Ter Steege et al. [37] noted that more than half of the roughly 5000 Amazonian tree species with valid species names occur in large-river floodplains. Compared to other freshwater wetlands on Earth, which usually consist of a handful of flood-tolerant tree species in comparison to richer upland forests, the exceptional tree species richness in Amazonian wetlands confirms that flooding or at least periodical waterlogging is a common stressor in Amazonian rainforests to which many tree species have adapted over evolutionary periods. Most of the potentially flood-adapted tree species of the Amazon have their distribution optima in the uplands and are only facultative colonizers of seasonally flooded environments, i.e., [5,38]. A significant part of floodplain tree species, however, have most of their populations in Amazonian river wetlands, and thus are "true" wetland specialists. For example, from the 2166 tree species with valid species names detected to occur in várzea floodplains, approximately 10% (207 tree species) were classified as hyperdominant and accounted for more than 50% of all their recorded stems in this ecosystem. Likewise, approximately 17% (139 tree species) of the 824 registered tree species with valid species names in igapó floodplains were classified as hyperdominants [37].

Gaining an integrated view of the floristic differences between várzea and igapó floodplains has been challenging, undoubtedly due to the vast size and heterogeneity of floodplains in the basin. Many transitional river wetlands types do not fit the classical categorization of várzea and igapó [39]. Furthermore, rivers cut across marked fertility, climatic, and diversity gradients across the Amazon basin. For example, there are many

river wetlands regarded as igapó in the western Amazon basin; however, their soils often consist of paleosediments from the Andes, and therefore are transitional and share tree species from both nutrient-rich and nutrient-poor environments, i.e., [40,41]. Because of the marked W–E fertility gradient, many wetlands classified as igapó in the western Amazon might have higher ecosystem fertility than várzea forests in the eastern Amazon basin. A better understanding of continental-wide species distribution patterns in large-river floodplains thus can only be achieved with more floristic inventories and improved classification systems of river wetlands and their ecosystem-level differences.

However, marked floristic differences between várzea and igapó floodplains occur at the regional level. Wittmann et al. [42] resumed the knowledge on floristic composition and diversity gradients in várzea and igapó floodplains, but since then, many more floristic inventories, particularly in under-sampled igapó forests, have become available.

### 3.1. Methods Applied in the Floristic Comparison

Using the database of the MAUA working group (Ecology, Monitoring and Sustainable use of Amazonian wetlands from the National Institute of Amazonian Research—INPA, Manaus-Brazil), we compare ~28 ha of floristic inventories performed in the central, northern, and western Brazilian Amazon in each of the ecosystems. The database consists mostly of published floristic inventories, but also includes seven so far unpublished ones along the Solimões and Japurá Rivers (Table 1). In the inventories, all tree individuals $\geq 10$ cm diameter at breast height (dbh) were recorded and identified to species through consultations of herbaria specimen and/or botanical experts. Because both várzea and igapó forests are characterized through marked species-richness gradients along the flood-level gradient, we only selected floristic inventories where the mean flood-duration data were available and relatively similar in both ecosystems (mean of $123 \pm 64$ days year$^{-1}$ in igapó and $130 \pm 65$ days year$^{-1}$ in várzea, respectively, Table 2). Flood duration in all plots was derived by calibrating water level marks on trees to the maximum water level recorded by the nearest river gauge during the sampling period (daily water-level measurements operated by the Brazilian Water Agency—ANA). For each inventoried tree, the watermark left on stems during the former high-water period was measured to the ground. With this reference point, the flooding history of each inventory was obtained by back-calculating hydrological data 30 years from the date of plot sampling [43].

**Table 1.** Metadata of the floristic inventories used for the comparison of Amazonian igapó (IG) and várzea (VZ) forests, including habitat type, river and geographic coordinates, plot size, plot mean flood period, number of inventoried individuals and species, Fisher's alpha diversity index, and publication authors. Total inventoried area sums up to 28.75 ha in igapó and 28.875 ha in várzea forests. CA = Central Brazilian Amazon, NA = Northern Brazilian Amazon, WA = Western Brazilian Amazon. * classified as paleo–várzea (black- or mixed-water rivers upon fertile (Andean) alluvial substrates).

| Habitat | Region/River | Lat./Long. | Size (ha) | Mean Flood Period (Days Year$^{-1}$) | No. ind. | No. Species | Fisher's Alpha | Author |
|---|---|---|---|---|---|---|---|---|
| IG | CA, Abacate River | −02.10, −58.43 | 1 | 70 | 747 | 59 | 15.02 | [44] |
| IG | CA, Abacate River | −02.09, −58.43 | 1 | 200 | 624 | 97 | 32.17 | [44] |
| IG | NA, Aracá River | −00.12, −63.29 | 1 | 52 | 1981 | 61 | 11.91 | [43] |
| IG | NA, Aracá River | −00.12, −63.29 | 0.5 | 68 | 865 | 51 | 11.85 | [43] |
| IG | CA, Cuyuni River | −00.42, −63.08 | 1 | 228 | 713 | 76 | 21.53 | [43] |
| IG | CA, Cuyuni River | −00.45, −63.10 | 0.5 | 122 | 982 | 36 | 7.34 | [43] |
| IG | CA, Jaú River | −01.83, −61.62 | 1 | 264 | 758 | 24 | 4.72 | [43] |
| IG | CA, Jaú River | −01.83, −61.61 | 1 | 216 | 609 | 46 | 11.55 | [43] |
| IG | CA, Jaú River | −01.83, −61.62 | 1 | 182 | 971 | 27 | 5.15 | [43] |
| IG | CA, Jaú River | −01.86, −61.59 | 1 | 191 | 758 | 55 | 13.63 | [43] |
| IG | CA, Jaú River | −01.86, −61.59 | 1 | 173 | 939 | 53 | 12.16 | [43] |
| IG | CA, Jaú River | −01.87, −61.59 | 1 | 124 | 986 | 54 | 12.28 | [43] |

**Table 1.** *Cont.*

| Habitat | Region/River | Lat./Long. | Size (ha) | Mean Flood Period (Days Year$^{-1}$) | No. ind. | No. Species | Fisher's Alpha | Author |
|---|---|---|---|---|---|---|---|---|
| IG | CA, Jaú River | −01.84, −61.59 | 1 | 101 | 450 | 72 | 24.2 | [43] |
| IG | CA, Jaú River | −01.90, −61.46 | 1 | 67 | 726 | 63 | 16.56 | [43] |
| IG | CA, Jaú River | −01.94, −61.44 | 1 | 53 | 674 | 69 | 19.25 | [43] |
| IG | CA, Cuieiras River | −02.36, −60.19 | 0.5 | 188 | 462 | 33 | 8.13 | [43] |
| IG | CA, Cuieiras River | −02.37, −60.19 | 0.5 | 75 | 525 | 29 | 6.6 | [43] |
| IG | CA, Uatumã River | −02.13, 59.03 | 1 | 128 | 459 | 110 | 45.85 | [45] |
| IG | CA, Uatumã River | −02.15, 59.03 | 1 | 270 | 548 | 28 | 6.24 | [45] |
| IG | CA, Negro River | −02.76, −60.76 | 1 | 92 | 468 | 78 | 26.72 | [46] |
| IG | CA, Negro River | −02.69, −60.78 | 1 | 92 | 398 | 51 | 15.54 | [46] |
| IG | NA, Negro River | −00.69, −63.16 | 1 | 85 | 722 | 62 | 16.24 | [46] |
| IG | NA, Negro River | −00.63, −63.26 | 1 | 86 | 815 | 53 | 12.68 | [46] |
| IG | NA, Jufaris River | −00.92, −62.29 | 1 | 91 | 593 | 57 | 15.54 | [46] |
| IG | NA, Jufaris River | −01.09, −62.05 | 0.75 | 52 | 851 | 65 | 16.37 | [46] |
| IG | NA, Padauari River | −00.15, −64.04 | 0.5 | 227 | 644 | 23 | 4.66 | [43] |
| IG | NA, Padauari River | −00.15, −64.05 | 0.5 | 112 | 270 | 69 | 29.94 | [43] |
| IG | NA, Padauari River | −00.15, −64.04 | 0.5 | 100 | 309 | 72 | 29.51 | [43] |
| IG | NA, Negro River | −00.24, −64.24 | 0.5 | 101 | 302 | 64 | 24.83 | [43] |
| IG | NA, Negro River | −00.24, −64.24 | 0.5 | 99 | 269 | 50 | 18.09 | [43] |
| IG | NA, Negro River | −00.21, −64.25 | 0.5 | 39 | 328 | 57 | 19.93 | [43] |
| IG | NA, Negro River | −00.35, −63.91 | 1 | 90 | 504 | 64 | 19.43 | [46] |
| IG | NA, Negro River | −00.45, −64.78 | 1 | 79 | 500 | 66 | 20.37 | [46] |
| IG | NA, Negro River | −00.35, −64.31 | 1 | 74 | 573 | 59 | 16.45 | [46] |
| VZ | CA, Solimões River | −02.56, −64.70 | 1 | 93 | 566 | 94 | 32.16 | Unpublished |
| VZ | CA, Solimões River | −02.56, −64.69 | 1 | 160 | 507 | 31 | 7.28 | Unpublished |
| VZ | CA, Solimões River | −02.55, −64.69 | 1 | 114 | 462 | 49 | 13.86 | Unpublished |
| VZ | CA, Solimões River | −03.34, −60.11 | 0.875 | 115 | 421 | 64 | 21 | [47] |
| VZ | WA, Juruá River | −03.23, −66.05 | 0.5 | 161 | 354 | 73 | 27.9 | [40] |
| VZ | WA, Juruá River | −03.21, −66.00 | 0.5 | 112 | 285 | 51 | 18.09 | [40] |
| VZ | WA, Juruá River | −03.44, −66.04 | 0.5 | 156 | 358 | 60 | 20.62 | [40] |
| VZ | WA, Juruá River | −03.20, −65.99 | 0.5 | 115 | 300 | 78 | 34.23 | [40] |
| VZ | WA, Juruá River | −03.20, −66.01 | 0.5 | 132 | 348 | 74 | 28.77 | [40] |
| VZ | WA, Juruá River | −03.35, −66.02 | 0.5 | 170 | 409 | 47 | 13.71 | [40] |
| VZ * | WA, Jutaí River | −03.32, −67.44 | 0.5 | 170 | 297 | 29 | 7.95 | [40] |
| VZ * | WA, Jutaí River | −03.36, −67.49 | 0.5 | 131 | 413 | 59 | 18.83 | [40] |
| VZ * | WA, Jutaí River | −03.39, −67.49 | 0.5 | 110 | 368 | 78 | 30.27 | [40] |
| VZ * | WA, Jutaí River | −03.37, −67.50 | 0.5 | 200 | 261 | 40 | 13.18 | [40] |
| VZ * | WA, Jutaí River | −03.33, −67.44 | 0.5 | 169 | 695 | 58 | 15.05 | [40] |
| VZ * | WA, Jutaí River | −03.37, −67.48 | 0.5 | 82 | 356 | 84 | 34.69 | [40] |
| VZ | CA, Solimões River | −03.25, −59.97 | 1 | 190 | 486 | 38 | 9.65 | [47] |
| VZ | WA, Japurá River | −02.85, −64.91 | 1 | 162 | 662 | 37 | 8.46 | [47] |
| VZ | WA, Japurá River | −02.89, −64.88 | 1 | 129 | 841 | 36 | 7.64 | [48] |
| VZ | WA, Japurá River | −02.90, −64.88 | 1 | 152 | 487 | 48 | 13.21 | [48] |
| VZ | WA, Japurá River | −02.85, −64.91 | 1 | 139 | 461 | 89 | 32.83 | [47] |
| VZ | WA, Japurá River | −02.85, −64.92 | 1 | 139 | 462 | 108 | 44.35 | [48] |
| VZ | WA, Japurá River | −02.79, −65.06 | 1 | 141 | 504 | 86 | 29.81 | [48] |
| VZ | WA, Japurá River | −02.83, −65.04 | 1 | 87 | 444 | 149 | 78.69 | [47] |
| VZ | CA, Purus River | −04.13, −61.88 | 1 | 70 | 542 | 103 | 37.68 | [40] |
| VZ | CA, Purus River | −04.28, −61.85 | 1 | 90 | 603 | 76 | 23 | [40] |
| VZ | CA, Purus River | −04.36, −61.91 | 1 | 160 | 731 | 56 | 14.12 | [40] |
| VZ | CA, Purus River | −04.25, −61.75 | 1 | 90 | 457 | 85 | 30.76 | [40] |

**Table 1.** *Cont.*

| Habitat | Region/River | Lat./Long. | Size (ha) | Mean Flood Period (Days Year$^{-1}$) | No. ind. | No. Species | Fisher's Alpha | Author |
|---|---|---|---|---|---|---|---|---|
| VZ | WA, Japurá River | −02.36, −65.47 | 1 | 227 | 486 | 78 | 26.25 | Unpublished |
| VZ | WA, Japurá River | −02.37, −65.47 | 1 | 166 | 607 | 71 | 20.85 | Unpublished |
| VZ | WA, Japurá River | −01.76, −65.76 | 1 | 49 | 618 | 71 | 20.71 | Unpublished |
| VZ | WA, Japurá River | −01.77, −65.76 | 1 | 140 | 481 | 63 | 19.38 | Unpublished |
| VZ * | WA, Tefé River | −04.13, −65.97 | 0.5 | 40 | 239 | 73 | 35.83 | [40] |
| VZ * | WA, Tefé River | −03.98, −65.01 | 0.5 | 31 | 320 | 69 | 27.03 | [40] |
| VZ * | WA, Tefé River | −04.12, −65.08 | 0.5 | 168 | 349 | 70 | 26.35 | [40] |
| VZ * | WA, Tefé River | −03.98, 65.01 | 0.5 | 187 | 303 | 59 | 21.86 | [40] |
| VZ * | WA, Tefé River | −04.17, −65.12 | 0.5 | 100 | 329 | 74 | 29.7 | [40] |
| VZ * | WA, Tefé River | −04.16, −65.01 | 0.5 | 118 | 281 | 62 | 24.61 | [40] |

**Table 2.** Comparative floristic inventories of trees ≥10 cm of diameter at breast height (dbh) in central Amazonian igapó and várzea forests, based on the floristic inventories presented in Table 1.

| | Igapó | Várzea | Total |
|---|---|---|---|
| No. of inventoried plots | 34 | 38 | 72 |
| Inventoried area (ha) | 28.75 | 28.875 | 57.625 |
| Mean flood duration (days year$^{-1}$) | 123 ± 64 | 130 ± 65 | |
| No. of individuals (total) | 22,323 | 17,093 | 39,416 |
| No. individuals ha$^{-1}$ (mean, SD) | 656.5 ± 310 | 449.8 ± 139 | |
| No. of species ha$^{-1}$ (mean, SD) | 56.9 ± 19.1 | 67.7 ± 23.2 | |
| Identified individuals (%, mean) | 77.79 | 87.1 | |
| No. of identified species (total) | 464 | 494 | 761 |
| Fisher's alpha (mean, SD) | 16.84 ± 8.83 | 24.22 ± 12.88 | |
| The 12 most important species account for (%, OIV) | 20.03 | 18.34 | |

For floristic comparison, we calculated the importance value index -IVI, [49]. for each species in each plot. The IVI incorporates relative frequency, relative abundance, and relative dominance (sum of the basal area of all individuals of a species in a plot divided by the total basal area of all trees in the plot), and species are thus equally weighted. Overall, the most important species were determined using the overall importance value (OIV), which represents the sum of the relative IVI (rIVI) and the relative frequency (rF) in all plots [1]. Alpha-diversity was quantified with Fisher's a-diversity coefficient, using all individuals and species per plot including distinguishable morphospecies. Floristic similarity (beta diversity) between plots was calculated using Sørensen's index -SI [50], but only with reference to the individuals with species-level taxonomic identifications. Thus, the similarity values must be considered as relative values.

*3.2. Results of the Floristic Comparison*

In total, 39,416 stems >10 cm dbh were inventoried, out of which 56,6% were recorded in igapó and 43,3% in várzea floodplains (Table 2). All inventoried stems belonged to 958 species when including distinguishable morphospecies (593 in igapó, 633 in várzea). After excluding the morphospecies, 761 species with valid names remained, out of which 464 were recorded in igapó and 494 were recorded in várzea (Table 2). A total of 394 tree species (51.8%) were shared between the ecosystems, while 297 tree species (39%) were exclusive to várzea and 267 tree species (35.1%) exclusive to igapó. Várzea plots were floristically more diverse (mean ± sd Fisher's alpha = 24.22 ± 12.88) than igapó plots (mean ± sd Fisher's alpha = 17.62 ± 8.8, Table 2). Only 38 tree species in igapó and

44 species in várzea accounted for more than 50% of all inventoried stems, respectively. Mean floristic similarity within the ecosystems was comparatively similar, at 17.62 ± 11.8% and 20.56 ± 11.3% for igapó and várzea, respectively. When comparing igapó to várzea plots, however, the mean floristic similarity amounted to only 9.4 ± 5.3% (only about one of ten inventoried tree species in a random plot in either várzea or igapó is shared between both ecosystems).

The 12 most important tree species accounted for ~20% of the overall importance in igapó and for ~18% of the overall importance in várzea (Table 3). Among the most important tree species, only two (*Pouteria elegans* (A. DC.) Baehni, and *Hevea spruceana* (Benth.) Müll.Arg.) were shared between both ecosystems. The striking difference in floristic composition is also evident at higher taxonomic groupings, e.g., the family level. The relative importance of the 15 most important families indicates, among others, Fabaceae, Lecythidaceae, Sapotaceae, and Chrysobalanaceae as more important families in igapó than in várzea, while Malvaceae, Annonaceae, Lauraceae, and Moraceae were more important families in várzea (Figure 1).

**Table 3.** Relative overall importance of the 12 most important tree species in central Amazonian igapó and várzea forests, based on the floristic inventories presented in Table 1.

| | | Igapó | | | | |
|---|---|---|---|---|---|---|
| Rank | Species | Abund. (Total) | Abund. (Rel.) | Dom. (Rel.) | Frequ. (Total) | Importance (Rel.) |
| 1 | *Pouteria elegans* (A. DC.) Baehni | 883 | 3.955 | 3.278 | 23 | 2.817 |
| 2 | *Tachigali venusta* Dwyer | 554 | 2.481 | 3.326 | 10 | 2.112 |
| 3 | *Amanoa oblongifolia* Müll. Arg. | 584 | 2.616 | 2.499 | 8 | 1.846 |
| 4 | *Leptobalanus apetalus* (E. Mey.) Sothers and Prance | 437 | 1.957 | 1.563 | 29 | 1.685 |
| 5 | *Hevea spruceana* (Benth.) Müll. Arg. | 445 | 1.993 | 2.378 | 12 | 1.669 |
| 6 | *Macrolobium acaciifolium* (Benth.) Benth. | 350 | 1.567 | 2.523 | 17 | 1.664 |
| 7 | *Hymenopus heteromorphus* (Benth.) Sothers and Prance | 463 | 2.074 | 1.688 | 20 | 1.607 |
| 8 | *Gustavia augusta* L. | 502 | 2.248 | 1.815 | 12 | 1.566 |
| 9 | *Duroia velutina* (Spruce ex Benth. and Hook f.) J. D. Hook. ex Schumann | 407 | 1.823 | 1.487 | 13 | 1.333 |
| 10 | *Swartzia polyphylla* DC. | 290 | 1.299 | 1.709 | 14 | 1.25 |
| 11 | *Swartzia racemosa* Benth. | 344 | 1.541 | 1.609 | 11 | 1.244 |
| 12 | *Aldina heterophylla* Spruce ex Benth. | 212 | 0.949 | 2.123 | 12 | 1.236 |
| Σ 13-593 | | 16,852 | 75.491 | 73.996 | | 79.967 |
| | | Várzea | | | | |
| 1 | *Pseudobombax munguba* (Mart.) Dugand | 577 | 3.375 | 4.823 | 22 | 3.027 |
| 2 | *Luehea cymulosa* Spruce ex Benth. | 514 | 3.007 | 3.241 | 16 | 2.296 |
| 3 | *Eschweilera albiflora* (DC.) Miers | 363 | 2.123 | 2.438 | 23 | 1.828 |
| 4 | *Pouteria elegans* (A. DC.) Baehni | 390 | 2.281 | 1.732 | 19 | 1.592 |
| 5 | *Virola surinamensis* (Rol. ex Rottb.) Warb. | 272 | 1.591 | 2.101 | 18 | 1.472 |
| 6 | *Cecropia latiloba* Miq. | 341 | 1.995 | 1.716 | 15 | 1.437 |
| 7 | *Handroanthus barbatus* (E. Mey.) Mattos | 267 | 1.562 | 1.678 | 13 | 1.254 |
| 8 | *Mabea nitida* Spruce ex Benth. | 263 | 1.539 | 1.286 | 16 | 1.155 |
| 9 | *Pterocarpus rohrii* Vahl | 243 | 1.421 | 1.215 | 19 | 1.133 |
| 10 | *Hevea spruceana* (Benth.) Müll. Arg. | 206 | 1.205 | 1.354 | 18 | 1.093 |
| 11 | *Hevea brasiliensis* (Willd. ex A. Juss.) Müll. Arg. | 213 | 1.246 | 1.245 | 17 | 1.057 |
| 12 | *Eschweilera ovalifolia* (DC.) Nied. | 173 | 1.012 | 1.333 | 17 | 1.009 |
| Σ 13-633 | | 13,271 | 77.64 | 75.834 | | 81.646 |

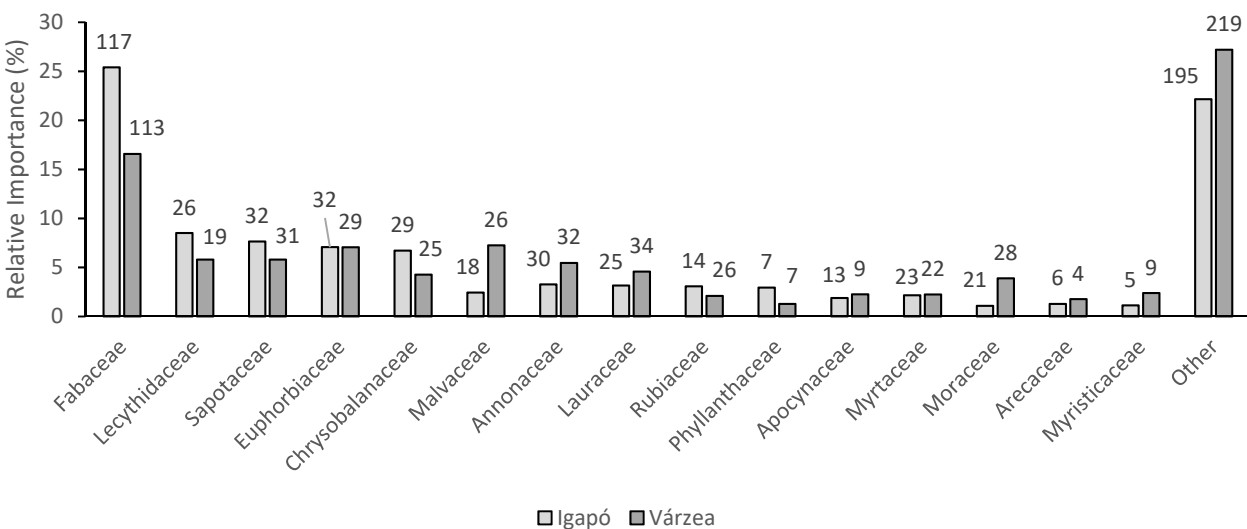

**Figure 1.** Relative overall importance of the 15 most important tree families in central Amazonian igapó and várzea forests (dbh ≥ 10 cm) based on the floristic inventories presented in Table 1. Numbers above bars represent total species numbers recorded in each family.

## 4. Ecosystem Specificity of Várzea and Igapó

### 4.1. The Role of Flooding

Flooding has similar effects on tree species distribution and diversity in várzea and igapó. One difference, however, is the flood-induced tree line that establishes at 9–9.5 m mean inundation height (mean flood duration of >300 days year$^{-1}$) in the central Amazonian igapó [51,52], whereas it is at approximately 7–7.5 (270 days year$^{-1}$) in the central Amazonian várzea [53]. The difference can be explained by the photon flux under water during the aquatic phase, which is higher in igapó (2–2.5 m) than in várzea (0.3–0.5 m), where tree seedlings and small trees are longer deprived of sunlight [42]. Possibly, periods of anoxia at the bottom of highly flooded igapó are also shorter than in várzea because of the smaller amount of easily degradable organic material, as a consequence of the low density of aquatic macrophytes in igapó [54].

In tropical regions, inundated soils turn conditions to hypoxic or anoxic within a few hours as a result of oxygen consumption by respiring roots and microorganisms, and the insufficient diffusion of oxygen through water and submerged tissues [55–58]. Oxygen depletion is accompanied by increased levels of $CO_2$, the anaerobic decomposition of organic matter, the increased solubility of mineral substances, and the reduction of the soil redox potential [59,60], followed by the accumulation of many potentially toxic compounds (e.g., $Fe^{2+}$, $Mn^{2+}$, $H_2S$), which is caused by alterations in the composition of the soil microflora [61]. Along sediment-laden rivers, sedimentation rates can be extreme and the deposits may additionally deteriorate soil aeration. In addition, the often high productivity of floating macrophytes in floodplains results in elevated decomposition rates, which further decrease the oxygen level [57]. Moreover, when flooding results in the complete submergence of trees, as, for example, in small individuals and seedlings, shoots are also deprived of sunlight [27,62], especially when floodwaters are poorly transparent.

Many Amazonian tree species combine several adaptive strategies to cope with stressful inundations at the level of roots and/or aboveground organs. Adaptations include partial or complete leaf shedding with the onset of flooding, the formation of adventitious roots and/or specialized roots such as pneumatophores, increased root biomass during flooding, the formation of hypertrophic lenticels on stems, the formation of root aerenchyma, the suberization of the root exodermis, the activation of fermentative enzymes under anaerobic conditions, as well as the production of elevated levels of antioxidant compounds, i.e., [27,63–70].

Tree species only establish on elevations where flood height and duration are tolerable [71,72]. Therefore, a clear zonation of tree species is evident along the flooding gradient [25,47,51,53,73]. Close to the flood-induced tree lines, monodominant forests develop [53,74,75]. With increasing topography and a reduced inundation period, tree species richness increases continuously towards unflooded forests of the uplands (terra firme) [1]. Most tree species clearly demonstrate a relatively narrow distributional niche along the flooding gradient. In an investigation of the niche properties of the most important tree species along the flood duration gradient, Householder et al. [43] stated that 73% out of 111 investigated tree species of the middle and lower Negro River floodplains occupied topographical niches that included less than 30% (85 flooded days) of the potential flood duration range. For Amazonian várzea, Marinho et al. [76] noted that four important timber tree species only successfully regenerated at significantly higher topographic positions than the positions of adult populations, indicating that the flood pulse affects species distributions especially during establishment and the early stages of life. Schöngart et al. [77] argued that years with exceptionally low water levels, as, for example, during El Niño years, expand the terrestrial period in floodplains and might be of crucial importance for the successful establishment of flood-sensitive tree species.

Specialized seed dispersal mechanisms are further adaptations in many Amazonian floodplain tree species [78,79]. Many floodplain tree species synchronize fruiting with high water levels [80–83], increasing the chances that seeds will be dispersed by water and aquatic organisms. The enhanced germination and growth rates of seeds or seedlings in dependence of its contact with river waters were reported in several floodplain tree species [79,84]. Many fruits and seeds are known for their air-containing tissues, which keep diaspores afloat for periods of up to two months [80,85–87]. Fish also may play an important role in seed dispersal and the reproductive dynamics of many Amazonian floodplain specialist tree species [21,88,89]. Several floodplain trees produce fleshy fruits that attract fruit-eating fish [85] and fruiting phenology is often synchronized with annual foraging patterns, thus increasing the chances of dispersal. Fish dispersal is potentially advantageous because it can move seeds against the prevailing water current and increase dispersal distance, especially in nonbuoyant seeds [90]. Furthermore, seed passage through the gut is thought to play a major role in breaking seed dormancy [87] and in increasing germination rates [91–93].

The development of specific adaptations to inundation is potentially accompanied by a tradeoff for life-history traits that make flood-adapted tree species less competitive in upland forests [42]. In a comparison of functional traits from tree species in flooded and unflooded central Amazonian forests, Fontes et al. [94] noted that flooded tree species had significantly larger leaf areas, wider vessels and higher xylem hydraulic conductivity than congeneric tree species of upland forests, while the upland trees had significantly higher wood specific density and lower stomatal density. This suggests that life-history traits in floodplain trees favor maximum hydraulic conductivity for the transport of water, oxygen, and nutrients for growth. Adaptation for fast growth is an advantage in floodplain environments where access to water and nutrients is generally high, but it also means that floodplain trees are more vulnerable to water stress as induced by climatic drought than upland trees, i.e., [95–97]. The cost of these adaptations is a loss of traits linked to tissue quality, resource conservation, protection against herbivores, mechanical strength, and long leaf life spans [98–101]. Taken together, life-history tradeoffs for floodplain survival reduces species competitive capacity in uplands. Wittmann et al. [5] investigated the spatioecological distribution of the 658 most abundant Amazonian várzea tree species and noted that 19–30% of the species showed significant preference for Neotropical floodplains, while approximately 10% of all investigated tree species were geographically and ecologically restricted to Amazonian várzea. Most endemic várzea tree species occurred in highly inundated low-várzea forest and here especially in the central Amazon, where flood amplitudes are high and flood durations longest [5]. Likewise, Amazonian igapó

forests are also thought to contain several endemic tree species [10,12,102], although no quantification of their numbers is available yet.

### 4.2. The Role of Nutrients

Due to their mineral composition and elevated fertility compared to large parts of central and eastern Amazonian uplands, Irion et al. [103] considered the Amazonian várzeas as "geochemical extensions of the Andes". The elevated fertility of várzea leads to high primary productivity of colonizing plants. Aquatic habitats are colonized by algae and aquatic and semiaquatic herbaceous species with exceptionally high productivity of up to 70–100 Mg ha$^{-1}$ year$^{-1}$ in the case of the semiaquatic grasses of the genera *Paspalum* and *Echinochloa* [104]. Progressively higher topographies are covered by different successional stages of forests, with the younger successional stages supporting among the highest forest aboveground net biomass primary productivity (31.8 Mg ha$^{-1}$ year$^{-1}$) reported on the globe [3].

Igapó soils are nutrient-poor, with up to 10- or more-fold lower content of elements such as P, K, Ca, and Mg when compared to várzea soils [11]. Semiaquatic herbaceous species are sparse and mostly consist of low-productive grasses (e.g., *Oryza* spp.) and sedges (Cyperaceae, [36]). Productivity of forests is low, as exemplified in the comparative measurements of diameter wood increments of several tree species that occur in várzea and igapó, including *Macrolobium acaciifolium* (Benth.) Benth. (Fabaceae), *Handroanthus barbatus* (E. Mey.) Mattos (Bignoniaceae), and *Vatairea guianensis* Aubl. (Fabaceae), which grow, on average, approximately two to five times slower in igapó than in várzea forests [77,105]. With a mean of 2.94 Mg C ha$^{-1}$ and year$^{-1}$, C-sequestration of igapó forest amounts to approximately 50% to that of Amazonian upland forests, and to approximately 30% to that of fast growing várzea forests [3,106].

The different nutrient levels of várzeas and igapós select for specific life-history traits important for the carbon- and nutrient-balance of colonizing tree species. Várzea tree species invest mainly in functional traits related to high resource acquisition and fast growth, while igapó species invest mainly in traits that allow for resource conservation and persistent tissues, as exemplified in the comparative measurement of functional traits of several congeneric tree species [107]. Differences in leaf attributes among várzea and igapó reflect strong growth-defense tradeoffs—while leaves are quickly produced and disposable in nutrient-rich várzea, leaves in igapó forests are longer lasting and better protected against herbivores [108–110] (Table 4). Many várzea tree species are deciduous during the flooded phase and/or renew leaves with the beginning of the terrestrial phase, while leaf deciduousness in igapó trees is less frequent [82]. Leaves in igapó trees are also usually smaller, vertically oriented, and often scleromorphic [45,111–113]. Conversely, leaves in várzea trees are N- and P-rich [114] and with higher photosynthetic capacities [27]. Differences in leaf attributes also affect decomposition rates—várzea leaves are usually quickly decomposed, and organic soil horizons are generally absent or sparsely developed, except when permanent inundation inhibits microbial activity (e.g., in chavascal sensu [53]). In contrast, igapó leaves need several years to decompose because of their comparatively high carbon content, leading to the development of thick litter layers [115,116].

The contrasting nutrient levels also influence fine root distribution, depth, and longevity. In igapó, fine roots are generally shorter, slower-growing, and longer-lived [117], and mostly develop as a thick superficial root mat, resulting in the efficient capture of scarce nutrients as they enter the substrate from the litter horizon. In várzea forests, fine roots are faster-growing and shorter-lived [117] and reach greater depths than in igapó (Table 4).

Lastly, tree reproductive strategies are also influenced by contrasting nutrient conditions. Many igapó tree species have large and heavy-weighted seeds, and seeds may remain attached to seedlings during several years, presumably delivering a nutrient supply and enabling seedlings to be less dependent on soil nutrients during the early stages of life [78]. Seeds of várzea trees are usually small, and the diaspores of many tree species,

and in particular those of highly flooded elevations, are dispersed by wind or water [118] (Table 4).

**Table 4.** Ecosystem characteristics of Amazonian igapó and várzea floodplains, important life-history traits of colonizing tree species, and most important biogeographic connections to other ecosystems. [1] [13], [2] [9], [3] [4], [4] [36], [5] [119], [6] [114], [7] [54], [8] [120], [9] [121], [10] [107], [11] [112], [12] [3], [13] [117], [14] [78], [15] [10], [16] [122], [17] [5].

| | Igapó | Várzea |
|---|---|---|
| Area [1] (km$^2$) | 302,000 | 456,000 |
| River origin [2,3] | Guiana and Central Brazilian Shields, pre-Cambrian and Paleozoic formations | Andes and Andean foothills |
| *Ecosystem characteristics* | | |
| Habitat stability | Stable during centuries to millennia [4] | Instable through sedimentation and erosion processes and channel migration [5] |
| Habitat diversity [5] | Low, mostly defined by flood-levels | High through small-scale hydrogeomorphic landforms and different flood levels |
| Fertility [6] | Low (intermediate in clear-water rivers) | High |
| Soil type [7] | Mostly arenosols with C content ~50% | Mostly different grained (sand-clay) Fluvisols and Inceptisols with C content ~40% |
| Risk of edaphic drought [8] | High, and very high at the highest flood levels | Low, but intermediate at the highest flood levels |
| Risk of fire [9] | High at the highest flood levels | Low |
| *Life-history traits* | | |
| Strategy [10] | Resource conservation | Resource acquisition |
| Leaves [11] | Small, hard, sclerophyllic, protected against herbivores | Soft, big, deciduous |
| Wood density [12] | Higher | Lower |
| Roots [13] | Surface or top 40 cm | 40–70 cm |
| Seeds [14] | Big and heavy, attached to seedlings | Small and light |
| *Biogeographic connections* | | |
| | Guiana and Central Brazilian Shields, Amazonian white-sand forests and savannas (campinaranas), Cerrado [15] | Western Amazon, Andes, Orinoco floodplains, Central American rainforest [16,17] |

### 4.3. The Role of Geomorphic Disturbance and Habitat Stability

Geomorphic disturbance as a result of fluvial dynamics is very distinct in Amazonian várzea and igapó floodplains, promoting an important ecosystem-level difference (Table 2). Most white-water rivers move their channels relatively quickly as a result of high sediment loads, comparatively low current velocities, and meandering activity. Next to the main river channels, sedimentation on point bars may reach up to 100 cm year$^{-1}$ [25,123]. Oppositely, on undercut slopes, lateral erosion locally may erode many hectares of forest during a single high-water period [53]. These processes result in channel migration that continuously destroys and recreates floodplain areas [119]. Rates of channel migration in Amazonian white-water rivers increase exponentially from the east to the west, ranging from 0.8% year$^{-1}$ at the Solimões River near Manaus [124] to 2.5% year$^{-1}$ at the Solimões River in western Brazilian Amazonia [125] to 14–23% along the Ucayali-Marañon in Peru [126]. Hydrogeomorphic dynamics are much less intense for both black- and clear-water rivers. Due to the small amount of suspended material, most river channels of black- and clear-water rivers are relatively stable over millennia [36,127]. Visible geomorphic changes along igapó riverbanks are restricted to relatively slow movements of channel bedload in mega-ripples [128] that eventually reshape sandy riverbanks when ripple edges are pushed against island and bank shorelines, as, for example, observed in some islands of the upper Mariuá archipelago (Brazilian Negro river).

High geomorphic dynamics along várzea rivers is a natural ecosystem disturbance that influences vegetation in important ways. At the landscape level, the continuous resetting of successional processes induces the creation of different-aged successional stages that usually coexist at small spatial scales (e.g., the same river section). The successional stages differ in forest structure (tree diameters and heights) and colonizing tree species, which dispose of different life-history traits related to establishment, productivity, and maximum tree ages [47,48,53,74]. Because the vegetation cover has feedbacks on geomorphological dynamics (e.g., by reducing water currents, promoting sedimentation, enhancing water

infiltration, etc., [53]), successional stages usually are good indicators for local environmental site conditions. Colonizing tree species of successional environments change along geographic gradients or among rivers, but their similar life-history traits and structural characteristics (e.g., maximum diameters and heights, diameter increment, and growth rates) enable the classification of forest types into pioneer-, secondary-, or late-successional várzea [1,53]. The interactions of várzea forest succession and local environmental site conditions are exemplified in Table 5.

**Table 5.** In Amazonian várzea next to the main river channels, the interaction of environmental and biotic variables generates a certain predictability of vegetation type distribution based on the classification in successional stages (sensu [129]). Exceptions in the progression may occur when highly flooded forest patches are subject to relatively slow sediment deposition rates, as exemplified in the successional model provided by Wittmann et al. [42]. [1] [53], [2] [130], [3] [3], [4] [47], [5] [68], [6] [104]. Table modified from [131].

| | Pioneer Stages | Early Secondary Stages | Late Secondary Stages | Late Stages |
|---|---|---|---|---|
| Environmental factors | | | | |
| Shear stress by hydraulic force | very high | high | intermediate | low |
| Mean inundation height (m) | 8–12 | 5–7 (230–270) | 3–5 (50–100) | <3 |
| Mean inundation duration (days year$^{-1}$) | >300 | 230–270 | 50–100 | <50 |
| Sedimentation rates | dm–m | cm–dm | mm–cm | mm |
| Substrate grain size [1] | sand | fine sand | silt | clay |
| Rel. PAR (%) [2] | 70–100 | 30–70 | 5–30 | <5 |
| Water retention capacity | low | very low | intermediate | high |
| Biotic factors | | | | |
| Vegetation type | macrophytes (i.e., semiaquatic grasses) | open shrub and tree formations | forest | forest |
| Tree density ha$^{-1}$ (> 10 cm dbh) [1] | - | 100–200 | 800–1000 | 400–600 |
| Mean tree diameters (cm) [3] | - | 10–15 | 15–30 | >30 |
| Tree heights in upper canopy (m) [1] | - | 8–10 | 15–20 | >20 |
| Stratification | - | single | double | double or more |
| Max tree ages (years) [3] | | 10–15 | 35–100 | >100 |
| Individual crown area (m$^2$) [4] | - | 30–60 | 60–200 | 200–800 |
| Aboveground root type [5] | | stilt roots | stilt and tabular roots | tabular roots |
| Wood density (g cm$^{-3}$) [3] | - | <0.4 | intermediate | >0.5 |
| Biomass (Mg ha$^{-1}$) | 70–100 [6] | 18 ± 3 [3] | 117 ± 9 [3] | 239 ± 11 [3] |
| NPP (Mg ha$^{-1}$ year$^{-1}$) [3,6] | 30–99 [6] | 11.25 [3] | 14.34 [3] | 6.46 [3] |
| Species richness (ha$^{-1}$) | 1–5 herb species | 1–3 shrub or tree species | 10–25 tree species | > 80 tree species |

The Amazon basin is usually densely forested and new site colonization is restricted to small gaps provided by tree fall or thunderstorm blow-downs [132], or to extreme environments, such as the savanna-like vegetation on white-sand soils (campinarana sensu [133]) and permanently inundated swamps (i.e., [134]). The constant hydrogeomorphic disturbance along white-water rivers and the provision of new substrates for the colonization by grasses (i.e., [69,135]), and subsequently by trees, occurs continuously at the landscape scale, and thus provides a unique opportunity for light-demanding pioneer species in the Amazon basin [34].

Many várzea tree species are well-adapted to hydrogeomorphic dynamics, in the sense that they are fast colonizers or regenerate well after disturbance. The colonizing species along highly disturbed riverbanks are pioneer species that combine a suite of

functional traits for different, potentially stressful conditions. First, these species are highly flood-tolerant. Second, pioneer tree species do not only tolerate, but demand for the high solar radiation incidence to be competitive. Likewise, they develop deep root systems of up to several meters to cope with shear stress induced by water currents and with drought conditions near the soil surface during the low-water stages [68]. Third, várzea pioneer species have to cope with sediment burial close to river channels, which covers fine root systems established near the soil surface, and thus severely aggravates water and nutrient uptake. Attributes that make várzea pioneer species effective colonizers include rigorous vegetative and sexual reproduction [64,136], continuous production of small and wind- or water-dispersed seeds [70,118], the ability to produce new fine-root layers above the annual deposits, stilt roots that increase aeration during inundation [68], low wood density, a short life cycle, and fast growth [74]. The stems and roots of pioneer tree species promote sedimentation by creating drag and reducing the energy of flowing water for carrying loads. As loads are deposited, the relative topographic position of vegetated stands increases [47,53]. Following the classic model of forest succession proposed by Connell and Slatyer [137], pioneer tree species subsequently shade their environment, inhibit the establishment of light-demanding grasses, and facilitate the establishment of other, moderately light-demanding tree species. After the establishment of early-secondary tree species, the regeneration of pioneer shrubs is inhibited because the understory light conditions below 30% relative photosynthetically active radiation (rPAR) are no longer suitable [130].

In low-dynamic igapó systems, successional processes are much slower. Reduced sedimentation rates allow the formation of closed-canopy forests down to the flood-induced tree lines. Shrubs and trees with classical pioneer characteristics (sensu [129]) are sparse, and if present at highly inundated sites, show growth rates which are 3–10-fold lower than that of their similarly flooded várzea counterparts. For example, the pioneer tree species *Symmeria paniculata* Benth. (Polygonaceae), *Malouetia* spp. (Apocynaceae), and *Eugenia* spp. (Myrtaceae), which are frequent along the black-water river banks of central Amazonia, reach individual ages of up to 100 years or more (J. Schöngart, unpublished data), while their white-water counterparts, such as *Salix martiana* Leyb. (Salicaceae), *Alchornea castaneifolia* (Humb. & Bonpl. Ex Willd.) A. Juss. (Euphorbiaceae), or *Tessaria integrifolia* Ruiz & Pav. (Asteraceae), complete their life cycles in 10–15 years [48,74]. While maximum tree ages reported for várzea late-successional stages hardly surpass 300 years [3,64,74,82], ages of up to 1000 years have been documented for highly flooded slow-growing *Eschweilera tenuifolia* (Lecythidaceae) in igapó [36,75].

Lastly, contrasting geomorphic disturbance regimes also cause important differences in habitat diversity between várzea and igapó (Table 4). Along white-water rivers, undercut and slip-off slopes change at relatively small spatial scales along river stretches, and erosion and sedimentation creates an alluvial landscape built up by point bars, islands, swales, ridges, secondary river channels, and lakes [119,138]. Small-scale differences in hydraulic regimes sort grains by size and weight and lead to substrate deposits of varying texture [53]. This leads to a high habitat diversity of different topographic elevations and textured soils in várzea floodplains at small spatial scales, which is especially pronounced in western Amazonia, where river dynamics are most intense [125,126]. In comparison, igapós consist of relatively few habitats [36] that are mostly defined by flood-level differences. The small amount of suspension load and the relative stability of river channels creates comparatively little environmental heterogeneity (Table 4). Islands are maybe formed in anabranching black- and clear-water rivers or along their braided stretches, such as in the Mariuá and Anavilhanas archipelagoes of the Negro River [127,139], but these also usually demonstrate little variation in topographies and substrate textures.

*4.4. The Role of Drought*

Drought is a major constraint for tree species distribution in large parts of the Amazon basin, in particular in its eastern part and towards the northern and southern transition to

the Neotropical savannas. From the west to the east, the number of consecutive months with less than 100 mm rainfall increases from 0 in the equatorial western part of the basin to up 7 in the eastern and southwestern part, along the transition to the Cerrado and Chaco, and towards the Venezuelan Llanos and parts of the Guiana Shield [140]. Drought conditions also depend strongly on the position of forests along the Amazonian flood wave, a seasonally migrating mass of water moving downslope that determines the magnitude and timing of local flooding. Forests of Amazonian large-river floodplains might experience low water levels at different periods of the year depending on their geographic location [15]. When low-water stages coincide with low precipitation during the dry season, drought can represent a severe stress for floodplain trees. Drought conditions also interact with soil texture, in particular when the poorly shaded parts of floodplains (e.g., the lower elevations) have sandy alluvial substrates that desiccate quickly. Drought impacts are especially reinforced during supra-annual El Niño events, when river water levels are lower than usual and the terrestrial phases in floodplains are expanded [141,142]. Keel and Prance [120] stated that drought may represent more limitations for several floodplain tree species survival than flooding. Most tree species have age-specific responses and different susceptibilities to environmental stressors [60], and it is thought that especially juvenile trees and seedlings might be most vulnerable to drought because their shallow root system adapted to flooding is inefficient for tapping deeper soil and ground waters, i.e., [61,143].

Amazonian tree species possess several adaptations for seasonal droughts at the physiological and morphological level [61]. Morphological adaptations include small, thick leaves with sclerophyllous structures and increased epicuticular waxes and hairs to reduce transpiration [144–146]. Leaf shedding, commonly interpreted as effective adaptation against anoxia during high water levels, is highest at the begin of the aquatic phase [82], although some species, in particular evergreen species, present a second, smaller peak of leaf exchange during the dry season [64]. Some Amazonian floodplain trees also respond to seasonal water shortage with sharply decreasing photosynthetic $CO_2$ assimilation [78,147,148] and a marked decrease in root respiration [149]. Because germination and establishment of floodplain trees is timed to the low-water stages, seeds germinate as soon as flood waters recede on moist alluvial substrates [26,79]. If water availability declines in upper soil layers, seedling establishment may be severely limited [150]. Consequently, seedling mortality of five investigated várzea tree species was significantly higher during the dry season than during the flooded period [70].

Although never directly compared at the ecosystem level, it is reasonable to expect that the role of drought has important regulating effects for tree species establishment and consequently for tree species distribution and floristic composition of várzea and igapó forests. Because of the higher amount of suspended material in river waters, várzea substrates may consist of a variety of soil grain sizes, but clay fractions increasingly dominate at higher topographical elevations and at sites more distant from the main river channels [53]. These clayey sites are usually densely forested and the amount of rPAR reaching the understory usually does not exceed 5% of the maximum possible value [130]. Under dense canopies, clayish substrates hardly desiccate, even during long-lasting dry seasons. Drought-induced limitations are thus mostly limited to the lower parts of floodplains, which have coarser substrates and less canopy coverage, where the rPAR reaching the ground ranges from 30 to 70% [130]. This situation is very different in igapó floodplains, where substrates form podzols that consist of up to 80% of fine sand. Higher parts of densely forested igapó substrates may accumulate shallow, silty to clayish topsoil horizons of some decimeters thickness [11,127]. However, these substrates desiccate much faster than those of the várzea. With an average of 8–15% of incoming rPAR (Wittmann, unpublished data), igapó forests also generally demonstrate higher levels of light incidence and lower relative air humidity at the forest floor, even in late-successional stages due to the lack of a dense subcanopy stratum [121,151]. Consequently, drought may be more pronounced in igapó than in várzea forests, which is reflected by the much higher frequency of drought-avoiding mechanisms in igapó trees [61,145,152] (Table 4).

### 4.5. The Role of Fire

Although most fires in the Amazon basin have anthropogenic origins [153,154], the occurrence of wildfires through lightning cannot be discounted as a selective force for floodplain tree species [155]. Igapó forests, and here especially the highly flooded elevations, are more frequently subject to fires than upland or várzea forests [121,156] because of a combination of several factors. First, the low water retention potential of sandy igapó substrates is especially prominent at the end of the dry season, particularly in El Niño years [155]. Second, the slow decomposition rates of leaves at highly flooded topographies lead to the accumulation of a thick litter layer. Combined with the fine root mats that establish at or near the soil surface, these factors provide a layer of fine fuel, which is about twice as large than in upland forests [116]. A comparatively low canopy and open understory characterizes particularly highly flooded topographies, where high radiation incidence results in a dry microclimate near the forest floor [151,155]. Once burned, the slow recovery times keep burnt areas open for several years, potentially exposing them to recurring fires, and the forests may be trapped relatively easily into a fire-dominated savanna state [156,157].

Carvalho et al. [155] investigated fire scars in igapó forests of the Jaú National Park (lower Negro River) over 35 years by remote-sensing techniques and found that 79% of the 254 detected fire scars occurred close (<10 km distance) to human settlements, suggesting that human activities are the main source of ignition. Over 90% of the burned area was associated to extreme hydroclimatic conditions during El Niño years. Because of the continuous increase of both more intense and more frequent drought events in large parts of the Amazon basin [158], fires might act as an increasing environmental filter for many fire-intolerant igapó tree species, particularly those of the highest flood levels [155].

### 4.6. The Role of Microclimates

Floodplains in general provide distinct microclimates when compared to adjacent uplands [159,160], but assessments of microclimatic differences among Amazonian floodplains are still lacking. Salati and Vose [161] described the water and energy cycle of the Amazon basin and noted that rivers often stand out as cloud-free areas due to the reduced latent and sensible heat distribution over the rivers as compared with that over upland forests. In general, the mosaic of aquatic and terrestrial habitats in floodplains causes an intense thermal heterogeneity in space and time, where nonvegetated substrates are characterized by most extreme diel thermal regimes, while aquatic habitats and dense floodplain forest exhibit attenuated thermal regimes [162,163]. The transpiration capacity of floodplain trees might be temporally higher than that of upland trees, but both flooding during the aquatic phase and increasing cumulative water deficit of soils towards the end of the dry season reduce transpiration rates significantly [26,164,165]. Altogether, Amazonian floodplains, and here particularly the less dense vegetated areas next to the main river channels, are subject to higher solar radiation incidence and higher temperature ranges than adjacent upland forests, while the amount of precipitation is likely reduced [166]. However, to what extent the attenuated latent heat capacity of water can buffer trees from both cold and heat [6,160] has never been tested in Amazonian floodplains. Likewise, the idea that naturally high groundwater levels in floodplains can buffer trees from drought [167,168] remains to be investigated. While the microclimates of floodplains clearly differ from those of the uplands, and also potentially between them, their specific role in determining species compositions in várzea and igapó are still largely unknown.

## 5. Biogeographic Patterns

### 5.1. Endemism and Habitat Specialization in Amazonian Floodplains

The existence of endemic tree species in freshwater wetlands is rare, if not absent, elsewhere, and points to the fact that flood-pulsing river systems in the Amazon basin exist since evolutionary time scales [5,42]. There is no clear paleo-environmental reason that drier climate conditions during past glacial periods (e.g., the last glacial maximum

at 18 thousand years BP) or wetter periods with increased sea levels at the interglacial maxima (e.g., the interglacial at 120 thousand years BP) interrupted the accumulation of wetland specialist tree species, although the river wetlands were incised and spatially reduced during drier periods and significantly enlarged during wetter periods [169,170]. Other large Neotropical freshwater wetland systems, such as the Pantanal [171], Orinoco Llanos [172,173], and riparian forests of the Cerrado and Atlantic rainforest [168] lack endemic freshwater tree species because the repeated dry and wet periods of the past likely repeatedly interrupted the evolution of flood-specific adaptations [171,174].

Flood adaptation is rarely mapped onto the phylogenies of Amazonian plant groups, meaning that most interpretations on this evolved from taxonomic information, largely from biogeographic comparison of species compositions of floodplains to other Amazonian and extra-Amazonian habitats. One hypothesis coming from this line of inquiry postulates that flood adaption has largely evolved from nonflooded terra firme. Kubitzki [12] stated that many Amazonian floodplain specialist tree species may have evolved sympatrically from the uplands as novel, flood-adapted ecotypes. Terborgh and Andresen [38] showed that family-level taxonomic composition of floodplain forests is more similar to nearby terra firme than other floodplain forest located elsewhere, implying a local origin of floodplain species from terra firme. Even at the species-level, the number of generalist tree species that floodplains have in common with non-flooded terra firme is high, especially where floods are short and ephemeral [1,5,12,38,175]. In the investigation of the distribution of várzea tree species, Wittmann et al. [5] found that 74.4% of the most important várzea tree species occur in Amazonian terra firme. The tremendous contact boundary that Amazonian floodplains share with adjacent uplands supports a great deal of species spillover. It is reasonable to expect that flooding is a strong selective pressure to which many Amazonian upland tree species are frequently exposed to. However, a key issue is how traits that confer flood tolerance might become fixed in floodplain populations when genetic spillover remains high with nearby terra firme populations. Nevertheless, in a rare molecular study addressing this question, populations of *Himatanthus sucuuba* (Spruce ex Müll. Arg.) Woodson (Apocynaceae) from floodplains and terra firme habitats were found to show strong genetic differentiation; however, these molecular differences were not accompanied by any taxonomically relevant morphological variation [176,177]. Far too few species have been similarly assessed as of yet, but the possibility that many cryptic floodplain species or races remain yet to be recognized is intriguing.

Because rainforests are characterized by high amounts of precipitation, it is reasonable to expect that many Amazonian upland tree species, from where most floodplain tree species likely originate, experienced local waterlogging in hydromorphic soils and/or through superficial inundations. Therefore, species already adapted to seasonal waterlogging, such as along episodically flooded riparian forests and associated swamps (in Brazilian Amazonian called "baixios") already possess a suite of traits that permit the colonization of the seasonally inundated Amazonian large-river floodplains. The "tree species colonization concept" in Amazonian large-river floodplains [42] rests on the assumption that many tree species experienced episodic, brief inundations in other Amazonian habitats, and "pre-adapted" over time to ecotypes that were able to colonize the higher topographies of flood pulsing river wetlands. Once adapted to the regular flood pulse, phylogenetic differentiation of tree species able to colonize the lower topographies of river floodplains was accompanied by a tradeoff for life-history traits that made flood-adapted tree species less competitive in the uplands. Due to the high number of floodplain specialists and even endemic tree species, we thus assume that the flood pulse is one of the most important drivers for sympatric speciation in the Amazon basin.

### 5.2. Floristic Links of Floodplain Floras within Amazonia and Beyond

Another hypothesis concerning the origins of flood adaptation in Amazonian plants posits that it is recruited from other biomes where the relevant adaptations already exist [178]. This view builds from the idea that the occupation of environmentally extreme

floodplains represents a large departure from the range of ecological strategies that have typically been successful within the predominant environmental condition in the Amazon—tall, closed-canopy, and non-flooded forest. Floodplains would offer extra-Amazonian taxa opportunity to track appropriate environmental conditions within a new Amazonian context.

Prance and Schubart [179], Prance [10], and Klinge and Medina [115] noted the floristic resemblance of the Amazonian igapó tree flora with the flora of Amazonian white-sand forests (campinarana sensu [133]). As with igapó, the woody vegetation of the campinaranas is often sclerophyllous and xeromorphic [111,180], generally species-poor in comparison to the adjacent terra firme forest, and composed of several endemic tree species [175,181–186]. Both ecosystems usually consist of many tree genera in common [186–188]. Environmental conditions shared between igapó and campinaranas include oligotrophic, nutrient-poor sandy soils with a low-water-retention capacity, higher than usual incoming solar radiation on the forest floor because of open canopies, and high amplitudes of oscillating groundwater levels between the dry and wet seasons that eventually shallowly flood campinaranas temporarily at the peak of the rainy seasons [45,110,184,188].

Igapó and campinaranas also share many tree species with montane forests of similarly nutrient-impoverished soils of the Guiana Shield [181,189,190]. In addition, a remarkable number of relatively important tree species of igapós is shared with both wetlands and uplands of extra-Amazonian savannas, such as the Cerrado (e.g., species of the genera *Calophyllum* (Calophyllaceae), *Tapirira* (Ancardiaceae), *Sacoglottis* (Humiriaceae), *Genipa* (Rubiaceae), *Handroanthus* (Bignoniaceae), *Amaioua* (Rubiaceae), *Senegalia* (Fabaceae), *Hymenaea* (Fabaceae) and *Tachigali* (Fabaceae)). Wittmann et al. [174] argue that species from these genera might effectively disperse along the riparian corridors that connect the Amazon with the Cerrado, such as the Araguaia-Tocantins, Xingu, and Tapajós rivers, while connectivity to the Guiana Shield is provided through rivers such as the Negro, Branco, Paru, and Araguari. Prance [10] and Kubitzki [12] argued that several taxa from savanna biomes are "pre-adapted" to seasonal flood pulses in igapós, proposing that the physiological response to seasonality in drought or flooding may be similar. This idea recognizes that convergent suites of attributes are often found in stressful environments, regardless of the particular environmental stressor, potentially allowing lineages to migrate more easily between biomes than between habitats within Amazonia.

Comparable floristic patterns exist in várzea. Preliminary data suggest that approximately 30% of the 658 most important várzea tree species have their distributional optima in cooler and wetter montane Andean forests (>1500 m) (F. Wittmann, unpublished data). For example, Andean and montane indicator genera such as *Cestrum* (Solanaceae), *Ilex* (Aquifoliaceae), *Hedyosmum* (Chloranthaceae), *Nectandra* (Lauraceae), *Salix* (Salicaceae), and *Tessaria* (Asteraceae) are extremely rare or absent in non-flooded Amazonian lowlands but can become dominant in floodplains [191]. Interestingly, similar results were reported for permanently inundated, oligotrophic Amazonian peat swamp forests, where Householder et al. [134] found that recent lowland peatland vegetation communities have taxonomic compositions appearing to be approximately $1050 \pm 391$ m above their actual elevations due to a high abundance and number of families with a high elevation optima. In both cases, the high moisture availability and heat-buffering capacity through water availability in wetlands might provide the optimal conditions for montane taxa to immigrate into the Amazon.

Species may not only be tracking the additional humidity found in floodplains, but also other favorable edaphic conditions. Wittmann et al. [5] noted that Amazonian várzea forests share up to 44% of all tree species with the Orinoco floodplains and 34% with the upland forests of Central America. Species shared to other Amazonian and extra-Amazonian ecosystems were comparatively low and amounted to approximately 20% for Amazonian igapó, 20% to the Atlantic rainforest, and 12% to uplands (e.g., Cerrado, Chaco) and hyperseasonal wetlands (e.g., Pantanal, Llanos de Moxos) of Neotropical savannas. That the Amazon and Orinoco floodplains share a large proportion of their tree species was formerly reported by Godoy et al. [122]. Most likely, the high floristic resemblance to

the Orinoco floodplains has its explanation by similar geoedaphic conditions of alluvial substrates that originate from the tropical Andes [1,122]. Likewise, central American rainforests are characterized by a large geodiversity, being partly characterized by nutrient-rich soils that derive from former and still-active volcanism, which could explain the disjunct distribution of many fertility-demanding várzea tree species with those of Central American rainforests.

## 5.3. Floodplain Ecosystems through Time

We have shown that igapó and várzea floodplains are characterized by contrasting environmental conditions and floras that indicate a very different evolutionary development. Igapó forests are clearly related to the flora of the Guiana Shield and the eastern parts of the Amazon basin. This "cratonic" flora is represented by a tree community where resource conservation strategies prevail [10,12,107,113]. In the uplands, eastern Amazonian forests are known for their comparatively tall trees, large basal areas, and high wood densities [2,192,193]. Databases on wood densities are still sparse in igapó, but community-level comparisons indicate higher mean wood density than in várzea, mainly due to the absence of fast-growing pioneer species in igapó [3]. Although igapó is restricted to cratonic Amazonia on the modern landscape, it is reasonable to expect that an igapó floodplain flora was more widespread and dominant before the main compressional uplift of the Andes during the Miocene [12].

With Andean uplift and the generation of new sediment in the western Amazon Pebas system, a novel type of relatively fertile ecosystem was introduced in the basin during the middle (20–10 Ma BP) to upper Miocene (10–7 Ma BP) [194,195]. Andean-associated tectonics and the sedimentary history of the Amazon basin generated broad-scale environmental and biological variation by the creation of a vast paleoalluvial template in one of the wettest biomes on Earth. This time period is interpreted as one of the most important periods of biotic diversification [33,196–198]. Most várzea tree species likely originate from this period, which was further accompanied by the collision of the North and South American plates and the intensified exchange of their floras, spurring diversification as well [33,191]. The várzea flora is clearly related to the western Amazonian alluvium with high fertility. Elevated ecosystem productivity leads to a community where resource acquisition strategies prevail [3,12,107]. Western Amazonian forests are usually more diverse than their eastern Amazonian counterparts [37,199], which is also reflected by higher tree species diversity in várzea compared to igapó [1,42]. In addition, western Amazonian forests are characterized by comparatively fast demographic traits and short turnover times that are related to higher diversification rates [200].

## 5.4. Floodplain Refugia

The widespread biogeographic connections of many floodplain taxa has led to the idea that rivers operate as important migration corridors through time and space. It has long been reported that otherwise drought-sensitive plants are often found along riparian zones in dry regions [167,168,201,202]. Examining the floodplain flora of the dry Cerrado region that links wetter Amazonian and Atlantic Forest biomes, Wittmann et al. [174] found that increased water availability in dry-area riparian zones accommodates a surprisingly rich community of drought-sensitive immigrant species from the wetter bordering biomes. This community is presumably tracking buffered wetland microclimates. In a space-for-time extension of this idea, Meave et al. [167] hypothesized that river floodplains could act as long-term drought refugia. If Amazonian large-river wetlands persisted even during adversely dry extremes, they likely represented important refuges for drought-sensitive tree species of the uplands, which possibly maintained populations near to the wetland–upland boundaries and/or the higher topographies in the floodplains, where access to groundwater was secured [5,34]. In the Amazon, large rivers are usually interpreted to be the divisors for a range of strictly terrestrial species, such as primates [203,204], lowland birds [205,206], lizards [207], butterflies [208], and even plant taxa [209]. In fact, the several

recognized areas or "districts of endemism" in the Amazon basin are sharply divided by major rivers, which might have favored vicariance in numerous terrestrial species groups [210]. However, for many tree taxa (virtually the 50% of all Amazonian tree taxa that occur in river floodplains), floodplain forests along all Amazonian large rivers likely have the opposite effect in attracting drought-sensitive tree species and providing tree refugia through time and space.

## 6. Conservation Implications

Brazil and other countries sharing the Amazon basin made important efforts in creating large networks of conservation units of different categories, which, together with indigenous lands, form large ecological corridors of protected areas. Meanwhile, more than 50% of the area of the Amazon basin is under protection, including large floodplain areas [211]. However, conservation designs must be improved by creating networks of protected areas and integrating floodplain corridors and their catchments [212]. Large complexes of the várzea floodplains are still not protected, especially in the Central and Eastern Amazon [213,214]. Furthermore, the categories and management plans of protected areas should be revised and adapted considering scientific knowledge, favoring strict protection of the vulnerable igapós and a moderate and sustainable use of natural resources in the more resilient várzea, which usually hosts a relatively high density of human populations due to its high productivity and richness in natural resources [215].

While these arguments are relatively well-known, here we want to call attention to the approximately 30% of floodplain specialist and endemic tree species of which Amazonian floodplain forests are composed. Most of these tree species have narrow distribution ranges along environmental gradients and thus are likely to be highly vulnerable to growing human expansion in and along Amazonian rivers, as well as to hydrological changes induced by river channeling, water deviation, and damming. Their importance in local floodplain forests also raises questions on the ecological functions they carry out. Ecological data regarding their interactions with terrestrial and aquatic biota is scarce, which is a regrettable situation as the flooding regimes that sustain them are increasingly imperiled. Floodplain forest degradation in Amazonian river floodplains is steadily increasing with the proliferation of hydroelectric dams and the alteration of downstream river hydrology that their operation causes [216]. Downstream impacts to floodplains may extend over 100 km of river stretches and associated floodplains and lead to decreasing floodplain tree diversity, the invasion of upland tree species, and large-scale diebacks of specialist floodplain trees [44,217–219]. The downstream spatial extent of these dam-induced impacts on biological communities has been coined as the "dam shadow" [219,220]. With 191 dams already in operation and >200 dams either planned or under construction, dam shadows may be triggering changes in Amazonian floodplain forest structure, composition, diversity, and function on large scales [219,221,222]. Furthermore, forest modification will cascade down to aquatic ecosystems, as they strongly depend on forest primary productivity [223,224]. As fish exploitation provides the main protein source and economic activity for ~80% of the Amazon's rural population [225], the degradation of floodplain forests will ultimately also have a direct and significant impact on human welfare in the region [16]. Amazonia's floodplain tree community is vital for ecosystem functioning and thrives in one of the most environmentally unique and severe hydrological regimes on the planet. Maintaining the hydrological integrity of these forests is paramount. With the threats now basin-wide and intensifying, the ecological consequences may be irreversible.

## 7. Conclusions

We have shown that Amazonian floodplain forests are important components of the Amazon basin that consist of specific, flood-tolerant tree floras. Igapós and várzeas have different floristic evolution, largely differ in their environmental settings, favor different life-history traits and ecological strategies of colonizing tree species, and have floristic connections to different ecosystems in the Amazon basin and beyond. These differences

create important spatial and temporal heterogeneity that sustain elevated levels of beta diversity. Because long-lasting and regular flood pulses act as important ecological filters and provide the evolution of flood-specific adaptations, Amazonian large-river floodplains likely also importantly contribute to sympatric speciation. In addition, floodplains provide tree refugia for drought-sensitive tree taxa of the surrounding uplands, a habitat function which is increasingly important in light of more frequent and intensified drought events in times of climate change [158,226,227]. Clearly, there are still many unanswered research questions on the ecology and evolution of the different Amazonian wetland floras which might be elucidated through an increasing number of phylogenetic studies in várzea, igapó, and terra firme tree species populations during the next decades. The distinction of Amazonian river wetlands into várzea and igapó is an important conceptual foundation that subsumes key biological and environmental aspects of Amazonian heterogeneity. Since the inception of this foundational concept nearly 70 years ago, it has illuminated our understanding of Amazonian ecology and evolution, figuring in influential models of species evolution, biogeography, and the maintenance of the biological diversity in the Amazon basin. This is a rich legacy that, in part, will also determine our capacity to protect Amazonian biodiversity in the near future.

**Author Contributions:** All authors contributed to the study conception and design. Material preparation, data collection and analysis were performed by F.W., J.E.H., L.O.D. and A.C.Q. The first draft of the manuscript was written by F.W., J.S., M.T.F.P. and W.J.J., and all authors commented on previous versions of the manuscript. All authors have read and agreed to the published version of the manuscript.

**Funding:** Floristic inventories were funded by the Brazilian Council of Science and Technology PRONEX-MCT/CNPq/FAPEAM "Tipologias alágaveis 2007", Universal (479599/2008-4), and CNPq-PELD 441811/2020-5, 441590/2016-0, and 403792/2012-6, INCT ADAPTA 465540/2014-7, CAPES 001, and FAPEAM Biodiversa 01.02.016301.03236/2021-74. Additional funding was provided by FIXAM/FAPEAM/017/2914 and CNPq grant 142200(2017-4), the ATTO Project (German Federal Ministry of Education and Research, BMBF funds 01LK1602F, and 01LK2101D, Brazilian Ministry of Science, Technology, Innovation and Communication; FINEP/MCTIC contract 01.11.01248.00), UEA and FAPEAM, LBA/INPA and SDS/CEUC/RDS-Uatumã, and the EU Project BiodivERsA—Clambio (BMBF 16LC2025A). We acknowledge support by the KIT-Publication Fund of the Karlsruhe Institute of Technology.

**Institutional Review Board Statement:** Not applicable.

**Data Availability Statement:** The data of published and unpublished floristic inventories are available at the MAUA-INPA data repository and can be solicited upon reasonable request.

**Conflicts of Interest:** The authors declare no conflict of interest.

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
