# Peer review of "A Review of the Ecological and Biogeographic Differences of Amazonian Floodplain Forests"

_water, doi:10.3390/w14213360_

Round 1

Reviewer 1 Report

The manuscript seems to be interesting. Although it is not a standard scientific paper (but only review-paper) I strongly recommend adding a short section Methodology to the manuscript. I suggest to Authors consider following comments: 1. A title should be changed (this is not biogeographical study, but review and current title is a bit confusing). 2. Section Abstract can be added by brief explanation why this paper fit the scope of the Sustainability journal - how exactly this paper can support sustainable management of Amazonian forests... In line 30 Authors say "We discuss", but section Discussion is missing in the manuscript. 3. Section Introduction needs corrections: Are there only two forest types (lines 53-54) or, in fact, these both two mentioned forest types (várzea and igapo) contain many different forest habitats by some classifications, It must be seriously explained. Secondly, definition of aim of the paper (lines 64-70) is very poor and must be seriously reworked as there is standard in review-papers. What is hypotheses? What new should bring this paper? Only description of a new published literature seems to be very poor aim (line 58). 4. Table 1 - how it was established? It needs more clarification in section Methodology (including explanation of using some unpublished data as Authors mentioned not very clearly in lines 65-66). 5. Section no. 4 Ecosystem specifity... deals with some ecological drivers for floodplain forest ecosystems, but there is missing explanation what drivers (and why) Authors considered (it should be explained in section Methodology). 6. I miss section Conclusion, where Authors briefly and precisely explain what are exactly original new findings of this review-paper and what should be studied in the future in order to fill revealed knowledge-gaps in the topic.   Finally, I do not like too much self-citations of the first Author in the section References, it shloud be shortened.   

Author Response

“I strongly recommend adding a short section Methodology to the manuscript.”

-Answer: We agree and assume that the reviewer mainly addresses the lack of a methodological description on the floristic comparison, which is the only “empirical” part of the study. Therefore, we decided to split topic 3. “Floristic differences between várzea and igapó” (L151 ff.) in 3.1 “Methods”(L189 ff.) and 3.2 “Results” (L223 ff.). In addition, we inserted two new Tables to the manuscript. Table 1(new) shows the metadata used for floristic comparison, and Table 3(new) lists the 12 most important tree species in each of the ecosystems for better comparison.

“A title should be changed (this is not biogeographical study, but review and current title is a bit confusing).”

-Answer: We agree, and changed the title clearly indicating “a review of…”

“Section Abstract can be added by brief explanation why this paper fit the scope of the Sustainability journal”

-Answer: We made no changes here, because the manuscript was (and is) submitted to “Water”, not to “Sustainability.”

“In line 30 Authors say "We discuss", but section Discussion is missing”

-Answer: we agree, and deleted the word.

“Are there only two forest types (lines 53-54) or, in fact, these both two mentioned forest types (várzea and igapo) contain many different forest habitats by some classifications, It must be seriously explained.”

-Answer: We made no substantial changes here. The review is on well-established ecosystems (citations L56 ff., L117 ff., etc.), which of course are composed of different forest types. We think that this is sufficiently described and clear throughout all sections of the paper. Different forest types (e.g., along the flooding gradient, or different successional stages) are intensely described in the sections 4.1 (flooding) and 4.3 (geomorphic disturbance), for example, but also in many other parts of the manuscript.

“Definition of aim of the paper (lines 64-70) is very poor and must be seriously reworked as there is standard in review-papers. What is hypotheses? What new should bring this paper? Only description of a new published literature seems to be very poor aim (line 58).”

-Answer: We agree. We were struggling with that but now we included specific research questions in the last paragraph of section 1. - Introduction. (L.65-74). With this, we think that the scope of the manuscript has much improved.

“Table 1 - how it was established? It needs more clarification in section Methodology (including explanation of using some unpublished data as Authors mentioned not very clearly in lines 65-66).”

-We agree. Methods were included in the text and also the metadata (including unpublished inventories) are now presented in Table 1

“Ecosystem specifity... deals with some ecological drivers for floodplain forest ecosystems, but there is missing explanation what drivers (and why) Authors considered (it should be explained in section Methodology)”

-Answer: we made no substantial change here. Our intention was to review the knowledge on the environmental differences between both ecosystems. Where environmental settings influence species composition (from what is known), the information was included in the topics 4.1 – 4.6.

“I miss section Conclusion, where Authors briefly and precisely explain what are exactly original new findings of this review-paper and what should be studied in the future in order to fill revealed knowledge-gaps in the topic. “

-Answer: We agree, and included the section 7. – Conclusions.

“Finally, I do not like too much self-citations of the first Author in the section References, it shloud be shortened.”

-Answer: We excluded two citations authored by the first author, which could be done because of double citations. However, much knowledge about these ecosystems was published by the first author, and any further cut of the reference list would imply in the omission of information.

Reviewer 2 Report

The manuscript “Biogeographic differences of Amazonian floodplain forestand their contribution to Amazonian tree diversity” by Wittman et al. compares two broad classes of Amazonian flooded forests. The manuscript provides a comprehensive survey of the literature and clearly summarises differences in the physical environments occupied by várzea and igapó forests. In terms of the tree flora the authors also summarise differences in the composition and diversity of these forest types.

The manuscript provides a thorough survey of the habitats and the physical drivers but I think more work is needed is around linking that to composition and diversity, the “biogeography” of these forests. This aspect is more poorly developed. Although, as the authors note, there are few evolutionary or biogeographical studies for individual taxa within these forests this does not mean there are not insights to be had (see next paragraph). Also I am not sure that the authors really address “their contribution to Amazonian tree diversity” as I do not see specific descriptions of this aspect.

In terms of explanations for the distribution of tree diversity the authors focus primarily on differences in physical environments (although they also mention the potential for river systems to act as dispersal corridors). The assumption seems to be that contemporary ecology is the sole driver the biogeography of Amazonian floodplain forest biogeography. Has this been shown empirically or are there any studies that invoke factors that occur over evolutionary timeframes? It would be worth at least considering the potential for evolutionary impacts.

While the text is generally well written there are some places where the current text is vague or confusing. I‘ve noted a couple of the more obvious ones below but there are others that while perhaps more minor would be worthwhile correcting. Having the manuscript by a native English-speaking colleague may help address this.

Ln 200-205 The graph does suggest that there are differences in the relative importance of various plant families in várzea and igapó forests. However, given the metric used and considering the description of diversity immediately before this (ln 180-199) it is difficult to evaluate just how important the differences are. For example, a relatively small subset of very common species (8-9% in each forest type) will dominate the metric such that seemingly substantial family level differences may be driven by a few species level difference.

Ln 613-616. There is no basis for this statement. As written it states that species accumulation occurred at the same rate despite environmental change over the last 120 thousand years. However, the cited references instead appear to describe landscape development and river dynamics not species diversification. Do the authors instead mean that that there is no clear paleo-environmental reason that species accumulation could not have occurred over the last 120 ky? This is quite a different observation.

Author Response

“The manuscript provides a thorough survey of the habitats and the physical drivers but I think more work is needed is around linking that to composition and diversity, the “biogeography” of these forests. This aspect is more poorly developed. Although, as the authors note, there are few evolutionary or biogeographical studies for individual taxa within these forests this does not mean there are not insights to be had (see next paragraph).”

-Answer: We agree. We elaborated research questions and included information about composition and diversity, e.g., by including a new Table 3. Wherever information was available about evolutionary or biogeographical studies, we included it in the sections 5.1. – 5.4.

“Also I am not sure that the authors really address “their contribution to Amazonian tree diversity” as I do not see specific descriptions of this aspect.”

-Answer: True. We changed the title and excluded this from the title and also the research questions.

“In terms of explanations for the distribution of tree diversity the authors focus primarily on differences in physical environments (although they also mention the potential for river systems to act as dispersal corridors). The assumption seems to be that contemporary ecology is the sole driver the biogeography of Amazonian floodplain forest biogeography. Has this been shown empirically or are there any studies that invoke factors that occur over evolutionary timeframes? It would be worth at least considering the potential for evolutionary impacts.”

Answer: The knowledge on evolution is resumed in section 5.3. To the best of our knowledge, there are no other publications describing factors that occur over evolutionary timeframes – and the time needed for the evolution of tropical tree species is also highly variable and largely unknown.

“While the text is generally well written there are some places where the current text is vague or confusing. I‘ve noted a couple of the more obvious ones below but there are others that while perhaps more minor would be worthwhile correcting. Having the manuscript by a native English-speaking colleague may help address this.”

Answer: Several parts of the manuscript were edited and/or rewritten by a scientist, native speaker, and co-author. It is difficult to correct specific “vague” parts, if these are not indicated or highlighted by the reviewer. 

“Ln 200-205 The graph does suggest that there are differences in the relative importance of various plant families in várzea and igapó forests. However, given the metric used and considering the description of diversity immediately before this (ln 180-199) it is difficult to evaluate just how important the differences are. For example, a relatively small subset of very common species (8-9% in each forest type) will dominate the metric such that seemingly substantial family level differences may be driven by a few species level difference.” 

Answer: Of course, we agree. The methods are now better described in section 3.1. In addition, we added species numbers per family in each of the ecosystems in the bars of Figure 1 to make the family-differences clear and comparable (in terms of species numbers). However, this is not a floristic study. Surely, many more analyses could be done in the floristic comparison, but our intension here was to provide an overview of the basic taxonomic differences, without too many details that would imply in many more pages of text. For a more detailed comparison, we will explore the floristic differences and publish it in an additional publication in future.

“Ln 613-616. There is no basis for this statement. As written it states that species accumulation occurred at the same rate despite environmental change over the last 120 thousand years. However, the cited references instead appear to describe landscape development and river dynamics not species diversification. Do the authors instead mean that that there is no clear paleo-environmental reason that species accumulation could not have occurred over the last 120 ky? This is quite a different observation.”

Answer: We agree, and changed the sentence according to this suggestion (now L670-676).

Round 2

Reviewer 1 Report

Authors made a serious effort in improvement of the former submitted manuscript. Now it is acceptable for publishing by my opinion.

Author Response

Thank you for the review. The suggestions helped in improving the manuscript.

Reviewer 2 Report

There remain various issues with the use of English. 

Para 1. "The considerable fine-scale habitat heterogeneity they generate..." The flooded forests themselves do not, strictly, generate habitat heterogeneity. Consider, instead "The considerable fine-scale habitat heterogeneity they contain..."

"because wetland occupation can allow organisms to tolerate larger temperature ranges as well as maintain more favorable edaphic and moisture conditions compared to adjacent, terrestrial habitats". As written the sentence implies that organisms maintain more favorable edaphic and moisture conditions; clearly these are habitat traits rather than organismal ones. The favorability or otherwise of wetlands is also species dependent rather than an absolute and this should be considered.

"In this sense, wetlands are likely to have conveyed spatial and temporal resistance and/or resilience to populations of species ..." Again, strictly nothing is "conveyed" to these populations. Consider, instead "In this sense, wetlands may allow for the survival of populations of species ..."

Para 2. The authors initially define the terms várzea and igapó with respect to a physical feature (i.e., floodplains). But emphasise the floristic differences of floras that occupy these areas. Do the terms apply to the forests or to the physical feature? Can the terms please be applied consistently throughout.

"..foundational for our understanding of the floristic differences and floral evolution of Amazonian floodplain forests" Do these papers cover evolution of the flower? The term "floral evolution" refers directly to evolution of the flowers. Is this what the authors had in mind. 

"While much of the knowledge resumed ..." The word "resumed" is used in both the English and Portuguese languages but differ in meaning. In this case the authors appear to be using the Portuguese ("summarised") rather than the English ("restarted"). Consider "summarised".

Para 3 Consider "What are the main environmental differences between these ecosystems?"

"Do the environmental differences track specific strategies and/or life-history traits of colonizing tree species?" This seems backwards the environmental difference won't "track" the biological features of the floral, instead we might expect plants with specific strategies or traits to occupy specific areas.

It is not possible to provide an exhaustive listing given the turn round time for this review but hopefully this highlights some of the issues.

With regard the summary of the data sets  it is indicated that there are 946 taxa (including morpho-species) yet based on the info provided in lines 238-241 the total taxa add to 958. Why the difference?

Author Response

Thank you for the second revision. We agree with all the suggestions. 

Our responses in detail:

Para 1. "The considerable fine-scale habitat heterogeneity they generate..." The flooded forests themselves do not, strictly, generate habitat heterogeneity. Consider, instead "The considerable fine-scale habitat heterogeneity they contain..."

We changed this sentence (L47-50).

"because wetland occupation can allow organisms to tolerate larger temperature ranges as well as maintain more favorable edaphic and moisture conditions compared to adjacent, terrestrial habitats". As written the sentence implies that organisms maintain more favorable edaphic and moisture conditions; clearly these are habitat traits rather than organismal ones. The favorability or otherwise of wetlands is also species dependent rather than an absolute and this should be considered.

We changed this sentence (L50-52).

"In this sense, wetlands are likely to have conveyed spatial and temporal resistance and/or resilience to populations of species ..." Again, strictly nothing is "conveyed" to these populations. Consider, instead "In this sense, wetlands may allow for the survival of populations of species ..."

We changed this sentence (L55).

Para 2. The authors initially define the terms várzea and igapó with respect to a physical feature (i.e., floodplains). But emphasise the floristic differences of floras that occupy these areas. Do the terms apply to the forests or to the physical feature? Can the terms please be applied consistently throughout.

We apply the terms consistently throughout, either as várzea and igapó (floodplain types) and/or as igapó and várzea forest (forest types).

"..foundational for our understanding of the floristic differences and floral evolution of Amazonian floodplain forests" Do these papers cover evolution of the flower? The term "floral evolution" refers directly to evolution of the flowers. Is this what the authors had in mind. 

We changed “floral” to “floristic” (i.e., L65, but also a few times subsequently)

"While much of the knowledge resumed ..." The word "resumed" is used in both the English and Portuguese languages but differ in meaning. In this case the authors appear to be using the Portuguese ("summarised") rather than the English ("restarted"). Consider "summarised".

Changed to “summarized” (L66)

Para 3 Consider "What are the main environmental differences between these ecosystems?"

Changed (L75)

"Do the environmental differences track specific strategies and/or life-history traits of colonizing tree species?" This seems backwards the environmental difference won't "track" the biological features of the floral, instead we might expect plants with specific strategies or traits to occupy specific areas.

We changed “track” by “favor” (L76, but also L946)

It is not possible to provide an exhaustive listing given the turn round time for this review but hopefully this highlights some of the issues.

With regard the summary of the data sets  it is indicated that there are 946 taxa (including morpho-species) yet based on the info provided in lines 238-241 the total taxa add to 958. Why the difference?

Indeed. The correct number is 958. We corrected that (L238).